# Clinical and Pathological Staging Discrepancies in Laryngeal Cancer: A Systematic Review

**DOI:** 10.3390/cancers17030455

**Published:** 2025-01-28

**Authors:** Giancarlo Pecorari, Andrea Lorenzi, Matteo Caria, Gian Marco Motatto, Giuseppe Riva

**Affiliations:** Division of Otorhinolaryngology, Department of Surgical Sciences, University of Turin, 10126 Turin, Italy; giancarlo.pecorari@unito.it (G.P.); andrea.lorenzi@unito.it (A.L.); matteo.caria@unito.it (M.C.); gmotatto@cittadellasalute.to.it (G.M.M.)

**Keywords:** laryngeal cancer, larynx, clinical staging, pathological staging, cTNM, pTNM, upstaging, downstaging, overstaging, understaging

## Abstract

Laryngeal squamous cell carcinoma (LSCC) remains one of the most challenging malignancies within the head and neck region. Clinical staging (cTNM) serves as the foundation for therapeutic decision-making. However, limitations in current imaging techniques often result in discrepancies between cTNM and pathological staging (pTNM). This systematic review aims to synthesize the existing literature on the correlation between the clinical and pathological staging of LSCC. The potential failure of clinical staging to detect subtle infiltration, such as tumor spread through the anterior commissure, involvement of the piriform sinus, or submucosal extension across the cricothyroid membrane, must be taken into account. If a subset of patients, particularly those initially staged as cT3, harbor more advanced disease (i.e., pT4a), non-surgical organ preservation strategies are unlikely to achieve meaningful long-term local control.

## 1. Introduction

Laryngeal squamous cell carcinoma (LSCC) is among the most frequent and challenging head and neck malignancies. Globally, it causes over 189,000 new cases and approximately 100,000 deaths annually [1]. Accurate staging is crucial for guiding treatment decisions, informing prognostic assessments, and counseling patients. The widely adopted TNM classification system, established by the American Joint Committee on Cancer (AJCC), delineates tumor extent (T), nodal involvement (N), and distant metastases (M). This framework influences whether patients pursue organ-preserving strategies or require more extensive surgical interventions. Clinical staging (cTNM), derived from physical examination, laryngoscopic evaluation, and radiological imaging, often serves as the cornerstone of therapeutic planning [2]. However, the limitations of current imaging modalities frequently lead to discrepancies between cTNM and the pathological staging (pTNM) determined postoperatively through detailed histopathologic analysis. Such discordances may result in under- or overtreatment, potentially affecting oncologic outcomes, recurrence patterns, and long-term survival [3].

Recent investigations have underscored the inadequacy of existing imaging techniques for accurately assessing tumor size, depth of invasion, and nodal metastases. Although the pTNM designation is considered the most accurate representation of disease burden, its reliance on surgical specimens restricts its utility to patients already selected for operative management. Consequently, the balance between accurate preoperative staging and the avoidance of both incomplete tumor control and unnecessary functional morbidity remains tenuous. Preliminary evidence suggests that discordances between cTNM and pTNM classifications are not uncommon, yet the true magnitude of this phenomenon, as well as its clinical significance and prognostic ramifications, remains incompletely characterized. Such inaccuracies in clinical staging may, for instance, lead to the selection of radiation therapy for a tumor that is actually more extensive than initially perceived—an approach that could compromise optimal local control and survival outcomes [4,5,6,7].

This systematic review aims to critically examine existing literature on the correlation between clinical and pathological staging of laryngeal carcinoma. In doing so, it seeks to quantify the extent of staging inaccuracies—emphasizing the potential for both under- and overstaging at the time of diagnosis—identify factors contributing to these discrepancies, and clarify their impact on therapeutic decision-making and patient outcomes.

## 2. Materials and Methods

This systematic review and pooled analysis was conducted in accordance with the Preferred Reporting Items for Systematic Reviews and Meta-Analyses (PRISMA) guidelines [8]. Every effort was made to ensure a transparent and reproducible methodology. The PRISMA 2020 checklist is disposable as Appendix A. The protocol of this systematic review was registered in PROSPERO (CRD42025632263).

### 2.1. Study Selection Criteria

Studies were considered eligible if they included adult patients (≥18 years) diagnosed with LSCC who underwent surgical management, allowing for both a preoperative clinical TNM and a postoperative pathological TNM staging assessment. The primary outcome of interest was to determine the degree of concordance or discordance between cTNM and pTNM, specifically highlighting the occurrence of both over- and understaging. Inclusion was restricted to retrospective case series and other observational studies that reported sufficient data to link individual patients’ cTNM and pTNM classifications. Studies published in English between January 2010 and October 2024 in peer-reviewed journals were included. Articles focusing solely on non-squamous cell carcinomas, those that did not provide direct comparisons between cTNM and pTNM, or those failing to report sufficient patient-level data were excluded. Incomplete reports, conference abstracts, and non-English publications were also excluded.

### 2.2. Literature Search and Screening

A systematic literature search was performed on 31 October 2024 using PubMed, Embase, Scopus, the Cochrane Library, and Web of Science. Search strategies combined relevant keywords related to LSCC, clinical staging, and pathological staging. The PubMed search strategy was as follows: “(laryn* AND (cancer OR carcinoma OR tumor OR neoplasm)) AND (laryngectomy OR cordectomy) AND (cT1 OR cT1a OR cT1b OR cT2 OR cT3 OR cT4 OR cT4a OR cT4b OR pT1 OR pT1a OR pT1b OR pT2 OR pT3 OR pT4 OR pT4a OR pT4b OR T1 OR T1a OR T1b OR T2 OR T3 OR T4 OR T4a OR T4b OR overstaging OR understaging OR upstaging OR downstaging) NOT (hypopharyn* AND (cancer OR carcinoma OR tumor OR neoplasm))”. After finalizing this initial search, the query was adapted for the other databases to maintain consistency in the search parameters. Following the initial database search, the titles and abstracts of all retrieved articles were screened for relevance (Phase 1). Studies deemed suitable were then assessed in full text to confirm their eligibility (Phase 2). Reference lists of included articles were also manually reviewed to identify additional eligible studies. Duplicate records were removed using reference management software.

### 2.3. Data Extraction

Data were extracted independently by two reviewers (AL, MC) using a standardized, piloted electronic form. The extracted variables included: first author’s name, year of publication, study design, country of origin, number of patients, and details of imaging modalities used for preoperative staging. Crucially, the presence of both cT and pT data for each patient was documented, along with any other staging parameters necessary to determine the level of concordance between the clinical and pathological assessments. Any discrepancies between reviewers in the data extraction process were resolved through discussion or consultation with a third reviewer (GR), as needed.

### 2.4. Risk of Bias Assessment

The risk of bias in all included studies was systematically evaluated using the Quality Assessment of Diagnostic Accuracy Studies 2 (QUADAS-2) tool, which is specifically designed to assess the quality of primary diagnostic accuracy studies [9]. QUADAS-2 examines four domains—patient selection, index test, reference standard, and patient flow/timing—and evaluates each domain for both risk of bias and concerns about applicability. Tailored signaling questions addressed specific study-design features that could introduce bias. Based on predefined criteria, each study was categorized as having a low, high, or unclear risk of bias. To ensure transparency and reproducibility, two independent reviewers (AL, MC) performed the assessments, and any discrepancies were resolved through consensus or consultation with a third reviewer (GR).

## 3. Results

### 3.1. Identification and Selection of Studies

A comprehensive overview of the study selection process is presented in Figure 1. The initial literature search identified 2515 potentially relevant articles. After removing duplicates and excluding records automatically deemed ineligible based on predefined screening criteria, 638 publications remained for initial evaluation. Titles and abstracts of these 638 records were reviewed, resulting in the exclusion of 208 articles that did not meet the inclusion parameters. Consequently, 432 articles proceeded to the full-text assessment stage, of which 31 studies were ultimately included in the review.

### 3.2. Quality Assessment

A summary of the risk of bias and applicability concerns, evaluated using the QUADAS-2 tool, is provided in Table 1. Overall, the assessment indicated that all included studies demonstrated high-quality levels.

### 3.3. Overview of Included Studies

A total of 31 studies met the inclusion criteria. These investigations were carried out in multiple countries, including Italy (10 studies) [5,14,18,20,21,24,25,26,29,31], France (4 studies) [7,27,28,36], the United States (4 studies) [6,12,15,19], Turkey (3 studies) [16,30,33], China (3 studies) [11,17,22], South Korea (2 studies) [32,34], and one study each from Egypt [4], Jordan [10], India [13], Vietnam [23], and Serbia [35]. The publications appeared between 2010 and 2024. Study periods varied widely, beginning as early as 1990 and extending through 2021, reflecting a broad temporal span of clinical experience and data collection.

Among the included studies, the methodological designs differed substantially. Twenty-seven studies were retrospective in nature, categorized as retrospective cohort studies, retrospective reviews, or case series with medical record reviews [5,6,7,10,11,12,13,15,16,17,18,19,20,21,22,24,25,26,27,28,29,30,31,32,33,34,36]. Four investigations were conducted prospectively, comprising prospective cohort or prospective case-control studies [4,14,23,35]. In terms of the reported level of evidence, the majority (26 studies) were classified as Level IV [5,7,10,11,12,13,15,16,17,18,19,20,21,22,24,25,26,27,28,29,30,31,32,33,34,36], while five studies were reported as Level III [4,6,14,23,35].

The characteristics of the enrolled patient populations varied substantially across the studies. Collectively, the 31 studies included a total of 7939 patients, with individual sample sizes ranging from as few as 8 to as many as 4554 participants, reflecting considerable heterogeneity in the study scale. Seven studies focused exclusively on early-stage tumors [15,21,23,26,28,29,30], while 10 examined advanced-stage cancers [4,12,16,17,18,20,22,24,27,31]. The remaining 14 studies encompassed patients across all clinical stages [5,6,7,10,11,13,14,19,25,32,33,34,35,36].

Table 2 provides a detailed summary of the authorship, publication years, countries of origin, patient cohorts, study durations, methodological designs, levels of evidence, and diagnostic modalities for each included study.

The relationship between clinical tumor staging and pathological tumor staging is summarized in Table 3. Of the 7939 patients included in the analysis, 6862 demonstrated complete concordance between cT and pT, yielding an overall concordance rate of approximately 86.43%. Among the remaining patients, 7.5% (592) were overstaged (pT < cT), while 6.1% (485) were understaged (pT > cT). Concordance rates varied by cT stage, with rates of 82.41%, 82.03%, 78.14%, and 89.64% observed for cT1, cT2, cT3, and cT4, respectively. A substantial proportion of non-concordant cases in the cT2 and cT3 stages were clinically understaged. Specifically, only 1.12% of cT2 cases were subsequently classified as pT1 upon definitive pathological examination, and only 4.45% of cT3 cases were reclassified as pT1 or pT2.

Notably, 17.42% of cases initially classified as cT3 were understaged, with pathological analysis following surgery revealing these tumors to be pT4a. The concordance rate was significantly higher for advanced-stage tumors (cT3/cT4) compared to early-stage tumors (*p* < 0.01, χ^2^ test).

## 4. Discussion

### 4.1. Highlights from the Systematic Review

This systematic review highlights a critical yet often underappreciated facet of LSCC management: the potential for meaningful discrepancies between clinical (cTNM) and pathological (pTNM) staging. Despite the centrality of accurate staging to treatment planning and prognosis, our findings underscore a persistent gap between what preoperative imaging and clinical assessments purport to show and the actual pathological extent of disease. This discrepancy has profound implications for patient selection, therapy allocation, and eventual oncologic outcomes, particularly as organ preservation strategies assume an increasingly prominent role in managing LSCC.

Celakovsky et al. (2017) reported that discordance between cTNM and pTNM staging significantly impacts prognosis, with patients experiencing shorter disease-free survival (DFS) and disease-specific survival (DSS) when pathological staging reveals a higher tumor stage than clinical staging [3]. These observations underscore the critical need for precise preoperative staging to avoid misclassification that could adversely affect treatment outcomes.

In this review, the concordance between clinical and pathological staging was approximately 82% for early-stage tumors (cT1 and cT2). Tumors staged as cT3 demonstrated a lower concordance rate (78%), whereas those classified as cT4 exhibited a higher rate (89%). Overall, the concordance rate was significantly greater for advanced-stage tumors (cT3 and cT4) compared to early-stage lesions, primarily driven by the high concordance observed for cT4 tumors. The notably lower concordance for cT3 lesions is particularly concerning; most discrepancies involved clinical understaging. Attention to this issue is warranted in therapeutic planning, as nearly one in five cT3 tumors may be more extensive than preoperative evaluations suggest.

### 4.2. Limitations of Current Imaging Modalities

A key factor contributing to staging inaccuracies lies in the inherent limitations of current imaging modalities. Preoperative imaging, including computed tomography (CT) and magnetic resonance imaging (MRI), is essential for staging LSCC and guiding therapeutic decisions. Contrast-enhanced CT is the primary imaging modality for evaluating LSCC due to its ability to avoid motion artifacts caused by breathing and swallowing, which are frequently encountered with MRI. Furthermore, CT facilitates the simultaneous staging of the thorax, making it a practical choice for comprehensive evaluation. Conversely, MRI has gained prominence as a complementary tool, and is particularly valued for its superior soft tissue contrast, which aids in detecting cartilage invasion and other subtle pathological changes that might be missed when using CT. Each modality has distinct advantages and limitations that influence staging accuracy [37].

Computed tomography demonstrates variable sensitivity and specificity in staging LSCC. Hartl et al. (2013) reported a sensitivity of 10.5% and a specificity of 94% for detecting thyroid cartilage invasion, with an overall accuracy of 87% [38]. Despite its utility, CT is prone to overstaging tumors involving the anterior commissure and understaging lesions associated with decreased vocal fold mobility. Similarly, Lim et al. (2011) observed an accuracy of 84.9%, with a high negative predictive value (95%), although the specificity dropped significantly in subsites such as the anterior commissure (42%) [39]. These findings highlight CT’s limitations in differentiating superficial from deeper invasions, particularly in anatomically complex regions.

Magnetic resonance imaging provides superior sensitivity for soft tissue evaluation compared to CT. Kuno et al. (2014) reported a sensitivity of up to 96% for detecting thyroid cartilage invasion, owing to MRI’s high contrast resolution. However, MRI’s specificity is lower, ranging from 56% to 65%, as inflammatory changes can mimic tumor invasion. Motion artifacts and reduced resolution in certain sequences further constrain MRI’s reliability [40]. Lim et al. (2011) highlighted MRI’s higher false-positive rate compared to CT, particularly in subsites like the anterior commissure, where the specificity was 65% [39]. Combining CT and MRI findings with clinical and histopathological data enhances the diagnostic accuracy and minimizes staging errors [41].

### 4.3. Clinical and Therapeutic Implications of Misclassification

These limitations can have outsized consequences for treatment planning. For example, submucosal infiltration along anatomically complex regions, such as the anterior commissure or the piriform sinus, or subtle vertical spread through the cricothyroid membrane, may go undetected. A tumor radiologically staged as cT2 may in fact be pT3 or pT4a once scrutinized microscopically. Conversely, what appears to be advanced infiltration might be downgraded upon pathological evaluation. Both scenarios risk suboptimal therapeutic decisions: the overtreatment of a patient with an overestimated T category can result in unnecessarily aggressive surgery and loss of laryngeal function, while the undertreatment of a patient with unrecognized deeper invasion may compromise oncologic control.

According to the NCCN Clinical Practice Guidelines in Oncology for glottic laryngeal cancer (version 1.2025) [42], for the early clinical stages (specifically cT1–cT2), both primary surgery and definitive radiation therapy (RT)—with or without a minor role for systemic therapy—are considered viable and equivalent therapeutic options. This is particularly true for cT1 disease, where well-established evidence and meta-analyses suggest that transoral resection and RT yield similar local control. Thus, at the cT1–cT2 level, accurate initial staging ensures that patients receive appropriate, less morbid treatments that may preserve organ function without an unnecessary escalation of therapy.

For more advanced-stage tumors such as cT3 LSCC, definitive concurrent systemic therapy/RT and surgery (often more extensive) assume roughly equivalent efficacy in terms of ultimate disease control, but the choice is influenced by the desire for organ preservation and patient-specific factors. Accurate staging at cT3 becomes critical: if a T3 tumor is overestimated as cT4a, the patient might be directed toward a total laryngectomy prematurely, foregoing the potential for organ preservation. Conversely, if a T3 lesion is underestimated and identified as cT2, the patient may not receive the intensified systemic therapy/RT or appropriately extensive surgical intervention needed to control a more advanced disease, thereby risking suboptimal oncologic outcomes.

At cT4a, the recommended initial approach is surgical (total laryngectomy with possible thyroidectomy and appropriate nodal dissection). In highly selected patients who decline surgery or are unfit for it, concurrent systemic therapy/RT or induction chemotherapy followed by RT can be considered, though these remain less standard options. Here, misclassification is particularly consequential: understaging a T4a tumor as cT3 could misguide the therapeutic strategy toward organ-preserving approaches that are less likely to achieve durable control. Overstaging a T3 as cT4a might unnecessarily push the patient toward total laryngectomy and associated morbidity, instead of attempting a more conservative yet still effective organ-preserving approach. Zbären et al. (1996) demonstrated that clinical and endoscopic evaluations alone often fail to accurately stage tumors, particularly in cases involving an invasion of the paraglottic and preepiglottic spaces or extralaryngeal tissues, with a clinical staging accuracy of only 57.5% [43]. This highlights the need for enhanced diagnostic precision to address the limitations of clinical assessment.

Over the past decades, the management of advanced LSCC has shifted dramatically. Early standards centered on total laryngectomy as the primary curative approach. Subsequently, seminal trials such as the Veterans Affairs study and Radiation Therapy Oncology Group (RTOG) 91–11 advocated for “organ preservation” via induction chemotherapy and/or concurrent chemoradiation (CCRT), reducing the immediate need for total laryngectomy [44]. However, these organ-preserving protocols did not necessarily guarantee functional preservation. As highlighted by Tomeh and Holsinger (2014), many patients treated with primary chemoradiation retain an anatomically intact larynx but suffer from persistent dysphagia, aspiration, or suboptimal voice quality [45].

Against this backdrop, the finding that clinical staging may fail to detect subtle infiltration—such as tumor spread through the anterior commissure, involvement of the piriform sinus, or submucosal extension across the cricothyroid membrane—takes on heightened significance. If a subset of patients, particularly those initially staged as cT3, in fact harbor more advanced disease (i.e., pT4a), non-surgical organ preservation therapies alone are unlikely to achieve meaningful long-term local control. In such scenarios, apparent “failures” of RT or chemoradiation may not stem from intrinsic radioresistance, but rather from an initially incorrect assumption regarding disease extent.

Moreover, the recognition that anatomical preservation does not always equal functional preservation has spurred renewed interest in conservative surgical approaches. Techniques such as open partial horizontal laryngectomy (OPHL) and transoral laser microsurgery (TOLM) can, in carefully selected patients, maintain physiologic swallowing, preserve acceptable voice quality, and minimize the need for permanent tracheostomies. It is equally important to consider that these surgical options can offer local control rates comparable or superior to those achieved with chemoradiation, provided that patients are accurately staged [26,46]. If imaging fails to identify subtle but significant submucosal infiltration or cartilage involvement, a patient deemed suitable for OPHL or TOLM may in fact require more extensive surgery [47].

### 4.4. Future Directions in Staging and Research

Beyond the technical shortcomings of imaging modalities themselves, a methodological critique that emerged from our review is the inconsistent reporting and definition of TNM categories. The AJCC guidelines emphasize the use of precise terminology to distinguish cTNM (based on clinical and radiologic findings) from pTNM (derived from pathological examination). Yet, many studies failed to adhere strictly to these labeling standards, often reporting only “TNM” without clarity on whether the classification was clinical or pathological. This lack of standardization hampers the ability to draw reliable comparisons across studies and complicates meta-analyses aimed at quantifying staging discrepancies. Moreover, Zbären et al. (1997) emphasized the therapeutic significance of detecting cartilage invasion, as its presence can preclude organ-preserving treatments and increase the likelihood of RT failure [48]. The accurate assessment of cartilage involvement is essential to guide appropriate treatment decisions and optimize oncologic outcomes.

Looking forward, these considerations underscore the importance of future prospective studies that incorporate advanced imaging modalities, novel biomarkers, and standardized endoscopic evaluations. This nuance is critical as treatment paradigms increasingly gravitate toward larynx-preserving strategies for advanced T categories. If a meaningful fraction of cT3 tumors are understaged pT4a lesions, the outcomes expected from chemoradiation protocols may be artificially skewed.

Additionally, closer collaboration between radiologists, pathologists, and head and neck surgeons will be essential. Multidisciplinary discussions that reconcile imaging findings with endoscopic and clinical data can help identify areas of potential underestimation. This approach aligns with the structured collaboration model proposed by Crosetti et al. (2024), emphasizing the integration of radiologic and surgical expertise for optimizing diagnostic precision and surgical planning in complex cases [49].

Technological innovations hold significant promise for addressing the persistent challenge of staging discrepancies in LSCC. Emerging tools such as artificial intelligence (AI) and machine learning (ML) offer a transformative opportunity to bridge this gap by providing objective, data-driven assessments that complement clinical expertise [50,51,52].

AI has already demonstrated substantial success in enhancing diagnostic accuracy in oncology. In LSCC, convolutional neural networks (CNNs) have been used to analyze complex radiological and endoscopic images, achieving sensitivities exceeding 90% in detecting tumor invasion and distinguishing benign from malignant lesions [51]. These tools excel in identifying subtle patterns of submucosal spread or cartilage invasion that may be overlooked during conventional clinical evaluation, addressing one of the critical sources of understaging in cTNM assessments. Additionally, an AI-enhanced integration of CT and MRI data has the potential to refine preoperative staging, offering more accurate predictions of pathological findings [53].

Beyond static imaging, AI-powered models that integrate multimodal data—such as radiologic imaging, endoscopic findings, and molecular biomarkers—may further enhance staging reliability. For example, incorporating histopathological data from biopsies into AI algorithms could help predict the likelihood of deeper infiltration or nodal involvement not evident from imaging alone [51,54].

However, realizing the full potential of AI in addressing cTNM-to-pTNM discrepancies requires robust validation. Prospective studies must evaluate these technologies against gold-standard pathological staging to ensure reliability and reproducibility. Furthermore, the standardization of AI algorithms and imaging protocols will be essential to facilitate their integration into clinical workflows [51,53].

Although the TNM classification remains a cornerstone of staging and prognostic assessment, growing evidence suggests that integrating molecular markers may enhance risk stratification and treatment selection in LSCC. Biomarkers such as cyclin D1, Ki-67, p53, and cadherin-1 have shown significant correlations with nodal metastasis, tumor grade, and overall survival, with elevated Ki-67 and downregulated cadherin-1 linked to more aggressive disease and higher TNM stages. Additionally, p53 mutations and Bcl-2 overexpression can predict poor responses to radiotherapy and chemotherapy, indicating a role for personalized therapeutic strategies [55,56,57]. Incorporating biomarker evaluations alongside TNM staging in future research and clinical practice could refine prognostic accuracy and inform targeted treatment approaches.

Such rigor in study design and data presentation will facilitate reliable comparisons across investigations, enabling meaningful meta-analyses that can synthesize heterogeneous findings into coherent evidence-based recommendations. In addition, enforcing strict adherence to AJCC staging criteria ensures that any observed discrepancies are less likely to stem from methodological inconsistencies, allowing researchers to focus on identifying intrinsic factors (e.g., tumor biology, imaging modality limitations) that contribute to misclassification.

Salvage surgery was not an exclusion criterion for this systematic review. Therefore, we also included data from studies about post-radiotherapy surgery in order to obtain a wider sample for pooled analysis. Moreover, the included studies did not always distinguish between naïve patients and irradiated ones. Further analyses that could distinguish these kinds of patients may be interesting. Lastly, a comparison between specialized academic institutions and general hospital practices may deepen the knowledge regarding the diagnostic accuracy of the tools used for clinical staging.

## 5. Conclusions

This systematic review underscores a fundamental challenge in LSCC management: the limited accuracy of clinical staging in capturing the true extent of disease. The inability of current imaging modalities to consistently detect subtle infiltration in critical anatomical regions leads to both over- and understaging, with significant clinical ramifications. For a meaningful proportion of patients, especially those considered for organ-preservation strategies, this inaccuracy may alter the trajectory of treatment decisions and outcomes. Future efforts must be directed toward refining imaging techniques, standardizing reporting, embracing multidisciplinary collaboration, and exploring novel diagnostic technologies. Only through such multifaceted initiatives can the field move closer to the ideal of perfectly aligning clinical expectations with pathological realities.

## Figures and Tables

**Figure 1 cancers-17-00455-f001:**
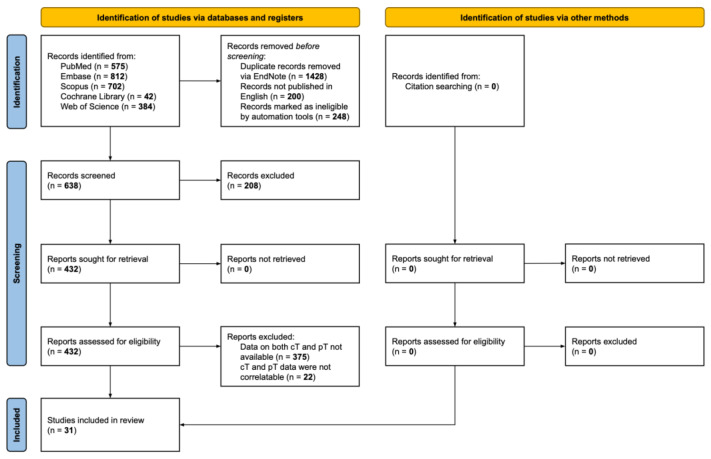
PRISMA (Preferred Reporting Items for Systematic Reviews and Meta-Analyses) flow chart of studies included in systematic review and pooled analysis. The review of the English literature was performed through PubMed, Embase, Scopus, Cochrane Library, and Web of Science, accessed on 31 October 2024.

**Table 1 cancers-17-00455-t001:** Risk of bias and applicability concerns for the included studies.

Author (Year)	Risk of Bias				Applicability Concerns		
	Patient Selection	Index Test	Reference Standard	Flow and Timing	Patient Selection	Index Test	Reference Standard
Tawfik et al. (2024) [4]	Low	Low	Low	Low	Low	Low	Low
Mohamad et al. (2023) [10]	Low	Low	Low	Low	Low	Low	Low
Zhao et al. (2022) [11]	Low	Low	Low	Low	Low	Low	Low
Frenkel et al. (2022) [12]	Low	Low	Low	Low	Low	Low	Low
Dwivedi et al. (2022) [13]	Low	Low	Low	Low	Low	Low	Low
Colizza et al. (2022) [14]	Low	Low	Low	Low	Low	Low	Low
Al-Qurayshi et al. (2022) [15]	Low	Low	Low	Low	Low	Low	Low
Tokat et al. (2021) [16]	Low	Low	Low	Low	Low	Low	Low
Contrera et al. (2021) [6]	Low	Low	Low	Low	Low	Low	Low
Benazzo et al. (2020) [5]	Low	Low	Low	Low	Low	Low	Low
Wang et al. (2019) [17]	Low	Low	Low	Low	Low	Low	Low
Ravanelli et al. (2019) [18]	Low	Low	Low	Low	Low	Low	Low
Mimica et al. (2019) [19]	Low	Low	Low	Low	Low	Low	Low
Locatello et al. (2019) [20]	Low	Low	Low	Low	Low	Low	Low
Lionello et al. (2019) [21]	Low	Low	Low	Low	Low	Low	Low
Zhang et al. (2018) [22]	Low	Low	Low	Low	Low	Low	Low
Tran et al. (2018) [23]	Low	Low	Low	Low	Low	Low	Low
Succo et al. (2018) [24]	Low	Low	Low	Low	Low	Low	Low
Carta et al. (2018) [25]	Low	Low	Low	Low	Low	Low	Low
Succo et al. (2016) [26]	Low	Low	Low	Low	Low	Low	Low
Gorphe et al. (2016) [27]	Low	Low	Low	Low	Low	Low	Low
Charbonnier et al. (2016) [28]	Low	Low	Low	Low	Low	Low	Low
Bertino et al. (2015) [29]	Low	Low	Low	Low	Low	Low	Low
Kayhan et al. (2014) [30]	Low	Low	Low	Low	Low	Low	Low
Mercante et al. (2013) [31]	Low	Low	Low	Low	Low	Low	Low
Han et al. (2013) [32]	Low	Low	Low	Low	Low	Low	Low
Topaloǧlu et al. (2012) [33]	Low	Low	Low	Low	Low	Low	Low
Foucher et al. (2012) [7]	Low	Low	Low	Low	Low	Low	Low
Cho et al. (2011) [34]	Low	Low	Low	Low	Low	Low	Low
Banko et al. (2011) [35]	Low	Low	Low	Low	Low	Low	Low
Hartl et al. (2010) [36]	Low	Low	Low	Low	Low	Low	Low

**Table 2 cancers-17-00455-t002:** Review of the English literature on clinical and pathological staging: included studies.

Author (Year)	Country	Patients	Study Period	Study Design	LoE	Diagnostic Modalities
Tawfik et al. (2024) [4]	Egypt	127	2016–2021	Prospective study	III	VL, CT, and MRI
Mohamad et al. (2023) [10]	Jordan	135	2008–2021	Retrospective review	IV	VL, MRI
Zhao et al. (2022) [11]	China	8	2018–2021	Retrospective cohort study	IV	N/R
Frenkel et al. (2022) [12]	USA	4554	2005–2016	Retrospective cohort study	IV	N/R
Dwivedi et al. (2022) [13]	India	30	2017–2021	Retrospective cohort study	IV	VL, CT
Colizza et al. (2022) [14]	Italy	17	2019–2021	Prospective case-control study	III	VL, CT
Al-Qurayshi et al. (2022) [15]	USA	214	2010–2016	Retrospective cohort study	IV	VL, CT
Tokat et al. (2021) [16]	Turkey	51	2013–2019	Retrospective cohort study	IV	VL, MRI
Contrera et al. (2021) [6]	USA	265	2001–2017	Retrospective cohort study	III	N/R
Benazzo et al. (2020) [5]	Italy	64	2015–2019	Retrospective review	IV	VL, CT
Wang et al. (2019) [17]	China	211	2007–2015	Retrospective cohort study	IV	VL, MRI
Ravanelli et al. (2019) [18]	Italy	17	N/R	Retrospective cohort study	IV	N/R
Mimica et al. (2019) [19]	USA	235	1999–2016	Retrospective cohort study	IV	VL (all), 23 CT, 1 MRI, 6 both
Locatello et al. (2019) [20]	Italy	30	2015–2018	Retrospective cohort study	IV	VL and, if needed, CT
Lionello et al. (2019) [21]	Italy	168	2000–2007	Retrospective cohort study	IV	VL, CT
Zhang et al. (2018) [22]	China	32	2000–2011	Retrospective cohort study	IV	VL, CT
Tran et al. (2018) [23]	Vietnam	32	2016–2017	Prospective study	III	VL and CT or MRI
Succo et al. (2018) [24]	Italy	479	2000–2012	Retrospective multi-institutional study	IV	VL and CT and/or MRI
Carta et al. (2018) [25]	Italy	17	2010–2017	Retrospective cohort study	IV	VL, CT
Succo et al. (2016) [26]	Italy	216	1995–2011	Retrospective cohort study	IV	VL and CT and/or MRI
Gorphe et al. (2016) [27]	France	29	2001–2013	Retrospective review	IV	VL, CT
Charbonnier et al. (2016) [28]	France	110	1990–2013	Retrospective review	IV	VL, CT
Bertino et al. (2015) [29]	Italy	166	2001–2013	Retrospective review	IV	VL, CT
Kayhan et al. (2014) [30]	Turkey	13	2010–2013	Retrospective review	IV	N/R
Mercante et al. (2013) [31]	Italy	32	2003–2012	Case series with medical record review	IV	N/R
Han et al. (2013) [32]	South Korea	32	1998–2010	Retrospective review	IV	VL, CT
Topaloǧlu et al. (2012) [33]	Turkey	44	2001–2009	Retrospective cohort study	IV	VL and, if needed, CT or MRI
Foucher et al. (2012) [7]	France	127	1998–2005	Retrospective review	IV	VL, CT
Cho et al. (2011) [34]	South Korea	92	1994–2006	Retrospective review	IV	VL, MRI
Banko et al. (2011) [35]	Serbia	34	2009–2010	Prospective study	III	VL and CT and/or MRI
Hartl et al. (2010) [36]	France	358	1992–2008	Retrospective review	IV	VL, CT

CT, computed tomography; LoE, level of evidence; MRI, magnetic resonance imaging; N/R, not reported; VL, videolaryngoscopy.

**Table 3 cancers-17-00455-t003:** Concordance and discrepancy between cT and pT stages (*n*, %).

	pT1	pT2	pT3	pT4	Total
**cT1**	698 (82.41%)	96 (11.33%)	27 (3.19%)	26 (3.07%)	847 (100%)
**cT2**	9 (1.12%)	662 (82.03%)	98 (12.14%)	38 (4.71%)	807 (100%)
**cT3**	5 (0.44%)	46 (4.01%)	897 (78.14%)	200 (17.42%)	1148 (100%)
**cT4**	1 (0.02%)	5 (0.10%)	526 (10.24%)	4605 (89.64%)	5137 (100%)
**Total**	713 (8.98%)	809 (10.19%)	1548 (19.50%)	4869 (61.33%)	7939 (100%)

## Data Availability

Not applicable.

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
