# Peer review of "Clinical and Pathological Staging Discrepancies in Laryngeal Cancer: A Systematic Review"

_cancers, 2025, doi:10.3390/cancers17030455_

Round 1

Reviewer 1 Report

Comments and Suggestions for Authors

This manuscript has an original analysis about differences between clinical and pathological staging in laryngeal cancer and a review of the consequences of this misclassification.

For a reviewer, it has two different approaches:

On the one hand, the PRISMA methodology evaluates using bibliography the dissociation between clinical and pathological TNM in laryngeal cancer. It is well and correctly developed.

In my opinion it is well done, but I miss a table that shows which diagnostic techniques are used for staging in the 31 selected studies (MRI, CT, PET, direct endoscopy, narrow band imaging (NBI)-enhanced endoscopy, perhaps indirect laryngoscopy in the older ones?) or at least detailing each technique in how many studies it is used.

I think that study from Mimica X et al. is not correctly included, as surgery is salvage treatment after radiotherapy.

On the other hand, the authors make an extensive discussion that seems based on opinions since many statements are made without support from references and provide little new knowledge.

For example:

-The entire discussion of therapeutic management is based on the NCCN guideline reference (for example, the statement that the indication for T4 is laryngectomy is not cited). In this part, in my opinion, the paragraph on lines 270-282 is not necessary, since the discussion on organ preservation protocols is not the focus of the study.

-From line 370 to 379 authors discuss research, but do not go into depth about whether some type of biological marker can provide more prognostic information than TNM itself.

-The whole discussion about AI has only one reference and I think it is a topic to develop further and perhaps needs a reference for all affirmations made from line 358 to 369 and specify if there is any specific aspect of the staging that already offers us improvement with AI.

-The discussion between lines 389 and 400, which deals with how to achieve a standardization of reports or data within a multidisciplinary team, is partly redundant and could be integrated into the paragraph on lines 340-350.

Focusing on some details of this:

From lines 270 to 307, as far as I understand, "implications for misclassification" are discussed, so it should be included in point 4.3. I think it would be more appropriate to put "highlights from systematic review", lines 259 to 269, at the beginning of the discussion. It may be more understandable to the reader

Line 274: a reference is required when Veterans Affairs Study is cited (if this paragraph is maintained)

Line 296: a reference is required when discussing partial laryngectomy

Line 343: a reference is required for Crosetti’s study.

This is a manuscript that can be published, but it should be made more understandable by simplifying the discussion, focusing on the implications of proper staging and supporting with references the key statements about the evidence in the management of laryngeal cancer.

Author Response

  • This manuscript has an original analysis about differences between clinical and pathological staging in laryngeal cancer and a review of the consequences of this misclassification. For a reviewer, it has two different approaches:

On the one hand, the PRISMA methodology evaluates using bibliography the dissociation between clinical and pathological TNM in laryngeal cancer. It is well and correctly developed.

In my opinion it is well done, but I miss a table that shows which diagnostic techniques are used for staging in the 31 selected studies (MRI, CT, PET, direct endoscopy, narrow band imaging (NBI)-enhanced endoscopy, perhaps indirect laryngoscopy in the older ones?) or at least detailing each technique in how many studies it is used.

  • Thanks for your comments and suggestions. A column with the diagnostic techniques used in the included studies was added to new table 2 (page 7). Moreover, QUADAS-2 was used for risk of bias and a new table 1 was added (page 5).

  • I think that study from Mimica X et al. is not correctly included, as surgery is salvage treatment after radiotherapy.
  • Salvage surgery was not an exclusion criterion for this systematic review. Therefore, we also included data from studies about post-radiotherapy surgery in order to obtain a wider sample for pooled analysis. However, distinguishing naïve patients from irradiated ones may be interesting and it was discussed (lines 496-501, page 14).

  • On the other hand, the authors make an extensive discussion that seems based on opinions since many statements are made without support from references and provide little new knowledge. For example:

The entire discussion of therapeutic management is based on the NCCN guideline reference (for example, the statement that the indication for T4 is laryngectomy is not cited). In this part, in my opinion, the paragraph on lines 270-282 is not necessary, since the discussion on organ preservation protocols is not the focus of the study.

  • Since we found that clinical T stage was not always reliable, it became crucial to be taken into consideration for non-surgical preservation protocols that were based on clinical T stage. Appropriate references were added.

  • From line 370 to 379 authors discuss research, but do not go into depth about whether some type of biological marker can provide more prognostic information than TNM itself.
  • We better described how biological markers can provide more prognostic information (page 13, lines 478-487).

  • The whole discussion about AI has only one reference and I think it is a topic to develop further and perhaps needs a reference for all affirmations made from line 358 to 369 and specify if there is any specific aspect of the staging that already offers us improvement with AI.
  • We added references about AI and we improved the discussion (pages 12-13, lines 431-466).

  • The discussion between lines 389 and 400, which deals with how to achieve a standardization of reports or data within a multidisciplinary team, is partly redundant and could be integrated into the paragraph on lines 340-350.
  • We integrated this part of the discussion in the previous paragraph to avoid redundancy.

  • Focusing on some details of this: From lines 270 to 307, as far as I understand, "implications for misclassification" are discussed, so it should be included in point 4.3. I think it would be more appropriate to put "highlights from systematic review", lines 259 to 269, at the beginning of the discussion. It may be more understandable to the reader
  • We modified the subtitles according to your suggestion.

  • Line 274: a reference is required when Veterans Affairs Study is cited (if this paragraph is maintained)
  • The reference was added.

  • Line 296: a reference is required when discussing partial laryngectomy
  • The reference was added.

  • Line 343: a reference is required for Crosetti’s study.
  • The reference was added.

  • This is a manuscript that can be published, but it should be made more understandable by simplifying the discussion, focusing on the implications of proper staging and supporting with references the key statements about the evidence in the management of laryngeal cancer.
  • According to your suggestions, we simplified the discussion, focusing on the implications of proper staging, and we supported it with more references.

Reviewer 2 Report

Comments and Suggestions for Authors

A retrospective analysis of data collected on clinical and pathological staging of laryngeal cancer patients.  The objective is to assess the consistency of the compatibility of pathologic and radiologic imaging assessment.

The following are main issues that  should be commented on to improve the manuscript.

1)     Comparison between academic  specialized institution and general hospital practices.

2)     Comment on data from the authors own institution.

3)     Modify the title to state “Radiological” imaging and pathological staging” instead of “clinical”.

Author Response

  • A retrospective analysis of data collected on clinical and pathological staging of laryngeal cancer patients. The objective is to assess the consistency of the compatibility of pathologic and radiologic imaging assessment.

The following are main issues that should be commented on to improve the manuscript.

1)     Comparison between academic specialized institution and general hospital practices.

  • Thanks for your comments and suggestions. The data extracted from the studies included in the review did not allow such comparison. However, it is interesting and it has been discussed (page 14, lines 501-503).

  • 2) Comment on data from the authors’ own institution.
  • This is a systematic review and we did not add any unpublished data from our institution.

  • 3) Modify the title to state “Radiological” imaging and pathological staging” instead of “clinical”.
  • We think that it is more correct to maintain the term “clinical”, instead of “radiological”, because it is the term officially used in the international TNM classification. Moreover, the clinical staging of laryngeal tumors is not only based on radiology, but also on endoscopy.